# A novel, rapid, quantitative method for face discrimination

**Kerri Walter**⊙*, Peter Bex

Northeastern University, Boston, MA, United States of America

* walter.ker@northeastern.edu

## Abstract

Face discrimination ability has been widely studied in psychology, however a self-administered, adaptive method has not yet been developed. In this series of studies, we utilize Foraging Interactive D-prime (FInD) in conjunction with the Basel Face Model to quantify thresholds of face discrimination ability both in-lab and remotely. In Experiment 1, we measured sensitivity to changes for all 199 structural Principal Components of the Basel Face Model and found observers were most sensitive to the first 10 components, so we focused on these for the remaining studies. In Experiment 2, we remotely investigated how thresholds varied when one component changed, compared to when two components changed in combination. Thresholds measured remotely were not significantly different from those measured in-lab (t(14) = 0.23, p = .821), and thresholds were significantly lower for components in combination than alone (t(7) = 2.90, p = .023), consistent with probability summation and Euclidean distance between faces, but not superadditivity. In Experiment 3, we replicated Experiment 2 with slight rotation to the faces to prevent pointwise comparisons. Thresholds were higher with rotation (t(30) = 4.32, p < .001) and for single than combined components, but did not reach significance (t(7) = 2.24, p = .061). Charts were measured in approximately 25.90 ± 8.10 seconds.

## Introduction

The ability to recognize and discriminate between faces is an important part of daily life (for recent review see [1, 2]). People with impaired facial recognition abilities report significant difficulties with social interaction and peer relationships [3, 4], as well as difficulties processing emotion [5], for review see [6]. Over the years there has been a growing literature on the behavioral indicators and neural patterns associated with face perception [7–9], however, although face processing has been widely and extensively studied, there is still no agreed method to quantitatively measure individual face discrimination ability.

### Face space models

Recent developments in computerized face analysis have led to the development of morphable face models [10–12] that enable continuous control of subtle differences in face properties. One database, the Basel Face Model, is a system from which high resolution 3D computerized

**Data Availability Statement:** All data and analysis files are available from the OSF data repository (https://osf.io/8wrvb/?view_only=66c2a0fb45a54ffca631c626831ddb72).

**Funding:** Supported by NIH R01 EY029713. The funders had no role in study design, data collection

and analysis, decision to publish, or preparation of the manuscript.

**Competing interests:** The FInD method is disclosed as patent (FInD) and held by Northeastern University, Boston, USA. FInD title: Method for visual function assessment; Application PCT/US2021/049250. PJB is a founder and shareholder of PerZeption Inc, which has an exclusive license agreement for FInD with Northeastern University. KW declares that she has no conflict of interest. This does not alter our adherence to PLOS ONE policies on sharing data and materials.

faces can be constructed and rendered at any angle, based on faces from a validated set of photographs that reflect the variations of emotion across the Big Five personality dimensions [10, 12–14]. The model was constructed from a principal component analysis [15–17] of 200 real adult faces, creating a model which contains 199 structural and textural Principal Components. In this way, the model creates a representative "face space", where an individual face is represented as a discrete point in the space and the dimensions are defined by the physical attributes of the source faces [18–20]. Thus, the closer two faces are located in the space, the more attributes they share and consequently, the more similar they appear. Previous authors have demonstrated that the space is isometric in that the relative Euclidean distance between faces represents the level of "different" that they appear, regardless of where in the space they fall [10, 12]. Recent researchers have examined the perceptual and computational distances between faces in the Basel Face Model and have reported that those distances can predict human performance when it comes to determining how similar two faces appear [21]. 3D morphable models such as this allow us to create realistic novel faces for which it is possible to specify and manipulate the ground truth characteristics [11, 12, 22, 23].

## Current face processing measurement models

Various tests exist to measure face processing ability, with prosopagnosia representing the lower end of the face processing spectrum. It is valuable to measure the entire spectrum of face processing ability, not just the lack thereof, as exemplified in The Oxford Face Matching Test (OFMT), introduced by Stantic et al. [24], which measures individual differences across the face processing ability spectrum, encompassing super recognizers, normal performers, and individuals with prosopagnosia. While prosopagnosia has relatively high prevalence, affecting up to 2.5% of the population [25, 26] depending on diagnostic criteria [27], its measurement is not an easy task. The diagnosis of prosopagnosia requires an impairment in face processing that is not attributable to low level visual deficits or general cognitive or object processing disorder [28]. However, some criticisms have been documented regarding the validity and analyses of some of these tests. Fysh and Ramon [29] observed that a patient with acquired prosopagnosia exhibited normal accuracy on multiple tests, including the Expertise in Facial Comparison Test (EFCT), Person Identification Challenge Test (PICT), Glasgow Face Matching Test (GFMT), and Kent Face Matching Test (KFMT). Rossion and Michel [30] emphasize the importance of evaluating reaction times and variability across items, as demonstrated in the Benton Facial Recognition Test (BFRT) [31]. Reaction times also serve as a critical component for diagnostic criteria in developmental prosopagnosia [32]. Additionally, the reliability for various face identity processing tasks is questionable, where participants are not necessarily consistent in performance across a variety of tests [33].

The two previously most commercially utilized tasks, the Benton Facial Recognition Test [31] and the Warrington Recognition Memory for Faces [34], while well suited for testing acquired prosopagnosia, have been questioned on their sensitivity for developmental prosopagnosia [35, 36] due to the ability to recognize clothing and hair from the test images and using simultaneous presentation and unlimited duration [37]. Since then, the Cambridge Face Memory Test [36] and Cambridge Face Perception Test [38] have been the standards for testing prosopagnosia. While these tests benefit from increased reliability, they still suffer from a lack of sensitivity to the degree of the face discrimination deficit or sensitivity to track gradual change due to the progression or remediation of prosopagnosia e.g. in autism [39–42], Turner's syndrome [43], schizophrenia [44, 45], Alzheimer's disease [46–48] or Parkinson's disease [49]. Questionnaires [50] can measure the participant's perceived impact of prosopagnosia but suffer from similar sensitivity limitations.

### Present study

In the present study, our aim is not to introduce an additional face identification diagnostic but rather to present a complementary paradigm that offers a quantitative measure of face processing ability. This approach not only provides an estimated threshold with very few trials but also ensures a fast and straightforward completion process.

We also wanted to investigate the face-space of the Basel Face Model in depth and did so by exploring all 199 structural components through a comprehensive component exploration. Through this we were able to create an optimized task using only the most salient features from the database. Using a condensed component sample allowed us to measure specific components individually and in combination, providing further insight on whether the features manipulated in the Basel Face Model are processed and perceived separately or in conjunction.

To address the limitations of current face perception tests, we developed a face discrimination paradigm that can be supervised or self-administered, in clinic or remotely and can be easily completed by clinical populations across a range of functional ability levels. We modified the FInD (Foraging Interactive D-prime, [51–53] paradigm to measure and quantify face discrimination thresholds in Face Space. FInD employs signal detection theory to estimate the difference between two stimuli required to reach a point of just noticeable difference, or in this case, the distance in face space at which two faces are perceived as two different individuals. FInD has previously demonstrated its versatility in measuring thresholds for color [54], contrast sensitivity [55], stereoacuity [56], and motion [51, 57]. Our adaptation of the paradigm to measure face space enhances its utility in advancing our understanding of face discrimination thresholds in a more comprehensive and quantitative manner. A patent for FInD protects commercialization of the method, but researchers are free and encouraged to use the method for non-profit purposes.

## Materials and methods

### Apparatus

Experiments 1 and 3 were run on a Dell Optiplex 7010 desktop computer (Dell Inc., Round Rock, TX) with an Intel HD Graphics 4000 graphics card (Intel Co., Santa Clara, CA). Stimuli were presented on a 32" LG Electronics Inc 32UD59 monitor (LG Electronics Inc., Seoul, South Korea) set to a screen resolution of 3840 x 2160 pixels at 30Hz. Participants were seated approximately 50cm from the screen. Experiment 2 was run on the experimenter's desktop which has an AMD Ryzen 5 1600 6-core processor (Advanced Micro Devices, Inc., Santa Clara, CA) and a NVIDIA GeForce GTX 1050 Ti graphics card (NVIDIA, Santa Clara, CA) and was viewed and controlled from the participants' own desktop or laptop, which was not calibrated. The experiments were programed using MATLAB (The MathWorks, Inc., Natick, MA) with Psychtoolbox [58]. Data was analyzed and boxplot figures were created in MATLAB using the Statistics and Machine Learning toolbox. This study's design and analysis were not pre-registered. All data and analysis code can be found at: https://osf.io/8wrvb/?view_only= 66c2a0fb45a54ffca631c626831ddb72.

### Stimuli

We utilized the Basel Face Model 2019 [13, 14] to construct unique and modifiable faces and to determine individual discrimination thresholds. The faces are specified by 199 different structural and textural component, each which controls a different aspect of the face. Each component varies in standard deviation units, centered on a mid-value of 0. Faces were

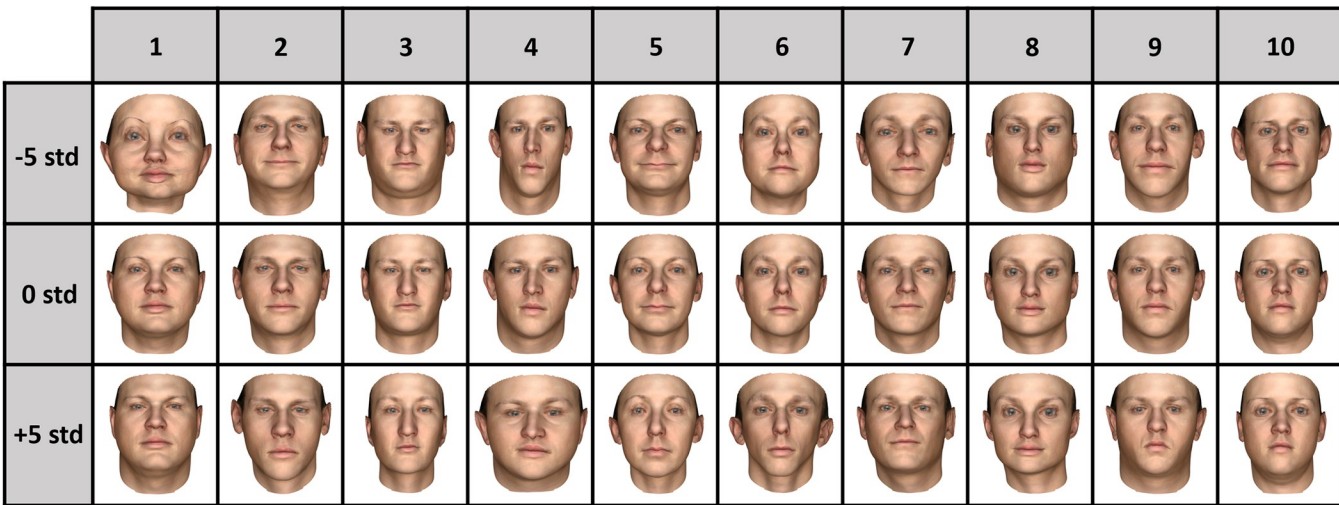

**Fig 1. Representative examples of the effect of the first 10 Basel Face Model components on face appearance.** Each column represents one component, where the middle row (0std) represents randomly generated faces, and the top and bottom rows show component values -5 and +5 standard deviations away from the 0std face, respectively. All texture components are 0, only structure components are altered. Republished from [14] under a CC BY license, with permission from Dr. Marcel Lüthi, original copyright 2018.

presented in pairs in each cell, the first face was generated with 199 random values (normally distributed random deviate with $\mu = 0$, $\sigma = 1.0$) for the structure and texture components. Separate random vectors for the 199 structural and 199 textural component coordinates were generated for each cell. The partner face in each cell was generated with the same component values as the first face except for the target component, which differed by an amount controlled by the FInD algorithm as described in *Procedure*, below. In other words, across face pairs, faces have different randomized vectors of 199 structural and 199 textural components, but within face pairs, faces share 198 structural components and 199 textural components.

In the first experiment, we measured face discrimination thresholds separately for all 199 components. The results showed that thresholds for the first 10 components were lowest overall (see Fig 3) and we therefore decided to study thresholds for only the first 10 Basel Face Model components in the second and third experiments. By changing one component at a time, we maintain specificity in threshold estimation, and by limiting the number of components tested, we maintain efficiency in threshold estimation. Representative examples of how faces are altered by each component are shown in Fig 1.

## Participants

Recruitment for this experiment took place between November 11th 2022 and January 27th 2023. For each experiment, we recruited 8 participants with normal or corrected-to-normal vision between the ages of 17 and 23 from Northeastern University's undergraduate population. All participants received psychology course credit as compensation for their participation. Participants (or guardians in the case of a minor) provided informed written consent on a consent form and demographic questionnaire approved by the institutional review board at Northeastern University before participation. Consent forms were completed before arrival, saved digitally, and verbally confirmed by the participant that they had completed the form upon arrival. The experimenter confirmed that the corresponding consent form was signed and cataloged before beginning the experiment. This totals 29 participants with demographic information (two participants repeated in Experiment 1, 1 author not included). For race, 14

participants reported White, 7 reported Asian, 4 reported Black or African American, 2 reported more than one race, 1 preferred not to say, and 1 reported unknown. Additionally, 3 reported Latino or Hispanic, and 1 reported unknown. For sex assigned at birth, 11 reported male, 18 reported female. As this is an exploratory study no estimates of variance are available to calculate sample size, therefore, based on an effect size of 1 standard deviation in a pilot study, we determined that we needed 8 participants for each experiment. These experiments were performed in accordance with the Declaration of Helsinki. IRB #: 14-09-16—Psychophysical Study of Visual Perception and Eye Movement Control.

## Procedure

We employed the FInD paradigm [51–53] to estimate discrimination thresholds. FInD is a self-administered paradigm in which stimuli are displayed over two or more charts (3 charts in Experiment 1, 4 charts in Experiments 2 and 3), each containing a grid of cells (here 3*3), a random subset (a uniform deviate between 0.6 and 0.8 of the total number of cells) of which contain a signal stimulus (2 different faces), the rest contain a null stimulus (2 identical faces), in random positions [51]. The observer's task is to identify with a mouse click, cells containing faces of different people. Each cell subtended 6° separated by a 3° gap. One target structural component of the Basel Face Model was varied per chart, the 199 components were investigated over multiple charts and participants. Each chart contained 9 face pairs. Each pair was generated with the same 199 random deviates ($\mu = 0$, $\sigma = 1.0$) for the structure and texture components, and different deviates were generated for each of the 9 face pairs on a chart. Thus, the two faces in a pair shared 198 structural and 199 textural components and appeared relatively similar, whereas the 9 face pairs differed from each other in all 199 structural and textural components and appeared relatively different from the other face pairs. The value of the target component in face pairs differed by a test level controlled by the FInD algorithm. Half of the test level was subtracted from the target component for one of the faces (left or right, at random), half of the test level was added to the target component for other face. For null face pairs, the value of the test level for null face pairs was 0, so all 199 components including the target component were identical. The placement of face pairs was random within the grid every trial. An example of a chart is illustrated in Fig 2.

Participants had unlimited time to click on cells that contained faces whose identity differed and not on cells where the face identities appeared to be the same. Once the participant had clicked a cell, the border was changed to green to indicate that the cell had been selected, and participants could click a cell an unlimited number of times to select or deselect it. Once they were satisfied with their selections, participants clicked on a 'next' icon to proceed to the next chart. The response in each cell was classified as a Hit, Miss, False Alarm or Correct Rejection, to calculate d' and the probability of a Yes response as a function of test level was calculated as:

$$p(Yes) = 1 - \Phi\left(\Phi^{-1}(1 - F) - \frac{d'_{max} \times \left(\frac{S}{\theta}\right)^{\gamma}}{\sqrt{\left(\left(d'_{max}\right)^2 - 1\right) + \left(\frac{S}{\theta}\right)^{2\gamma}}}\right) \qquad [1]$$

where p(Yes) is the probability of a Yes response, $\phi$ is the normal cumulative distribution function, F is the false alarm rate, S is the test level for the target component, $\theta$ is threshold level for the target component, $d'_{max}$ is the saturating value of d' which was fixed at 5, and $\gamma$ is the slope. This equation is a saturating function of d' as a function of stimulus intensity, based on computational [59] and physiological [60] studies; followed by a decision stage [61, 62]. Eq 1 is used in Matlab's *fittype()* function to create a specialized model which can then be used in Matlab's *fit()* function. We utilize the default *fit()* function with the default settings, which is a

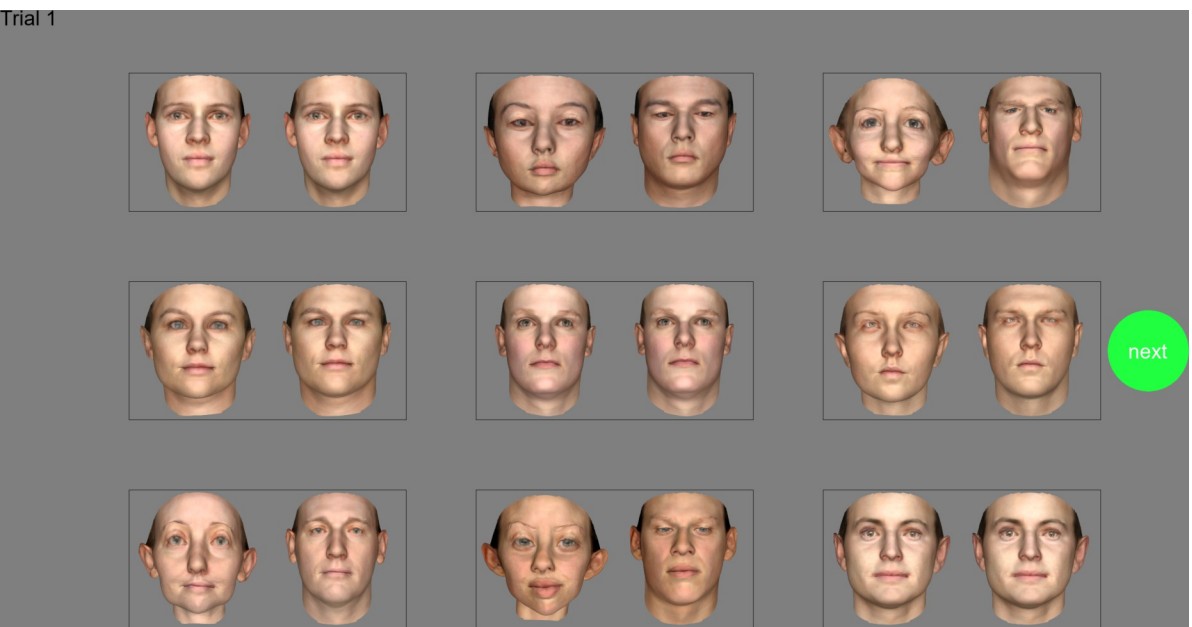

**Fig 2. Experimental procedure.** Example of one chart (in this case, component #1). In random locations, 3 cells are null and contain identical faces, the remaining cells are targets that contain faces in which the target component differs in logarithmically spaced steps, under the control of a FInD algorithm. Observers click on any cells that contain differing faces, then click on the 'next' button to proceed to the next chart. Republished from [14] under a CC BY license, with permission from Dr. Marcel Lüthi, original copyright 2018.

non-linear least squares method. The raw data from all charts are fit with Eq 1 after each chart using Matlab's *fit()* function. Observation weighting is applied to the fit as *1 / error estimate*. The fit to data from all completed charts was used to select the individualized range of face discrimination thresholds for each Basel Face Model component (from *d'* = 0.1 to 4.5) for face pairs on subsequent charts. The threshold difference for the target component of target face pairs on each chart (a random number between 0.6–0.8 of the total number of face pairs in the chart, here, 5, 6 or 7 target faces and 4, 3 or 2 null faces, respectively) was equally spaced between *d'* = 0.1 (difficult to detect) to 4.5 (easy to detect), as a personalized difference for each participant on the 2nd and subsequent charts.

**Experiment 1: Preliminary component exploration.** We first ran an in-lab study to measure discrimination thresholds for all 199 components to explore sensitivity throughout the Basel Face Model space. 7 naïve participants (2 male) and 1 author (KW) each measured face discrimination thresholds for 50 of the 199 components. There were 2 participants for each group of 50: 1–50, 51–100, 101–150, and 151–199 (this group completed 49). Thresholds for each component were measured in 3 charts. Charts were completed in an average of 24.24 ± 5.24 seconds. Fig 3A shows thresholds (the standard deviation difference for the target component) at which the faces appear different for each of the 199 components. components 1–50 had the lowest thresholds (mean thresholds for components 1–50 = 6.86; 51–100 = 9.96; 101–150 = 14.29; 151:199 = 14.67). To confirm these results, we measured thresholds for components 1–50 in 6 additional participants (for a total of 8 participants for components 1–50). 2 participants did not complete component 50 due to a programming error, which was corrected. Charts were completed in an average of 25.06 ± 6.00 seconds. Fig 3B shows thresholds for 8 observers (2 reproduced from Fig 3A), which are in good agreement with the preliminary study (F(1,6) = 0.52, p = .450, $\eta^2$ = 0.08), and thresholds were lowest for components 1–10 (mean thresholds for components 1:10 = 3.76, 11:20 = 6.30, 21:30 = 9.07, 31:40 = 9.94,

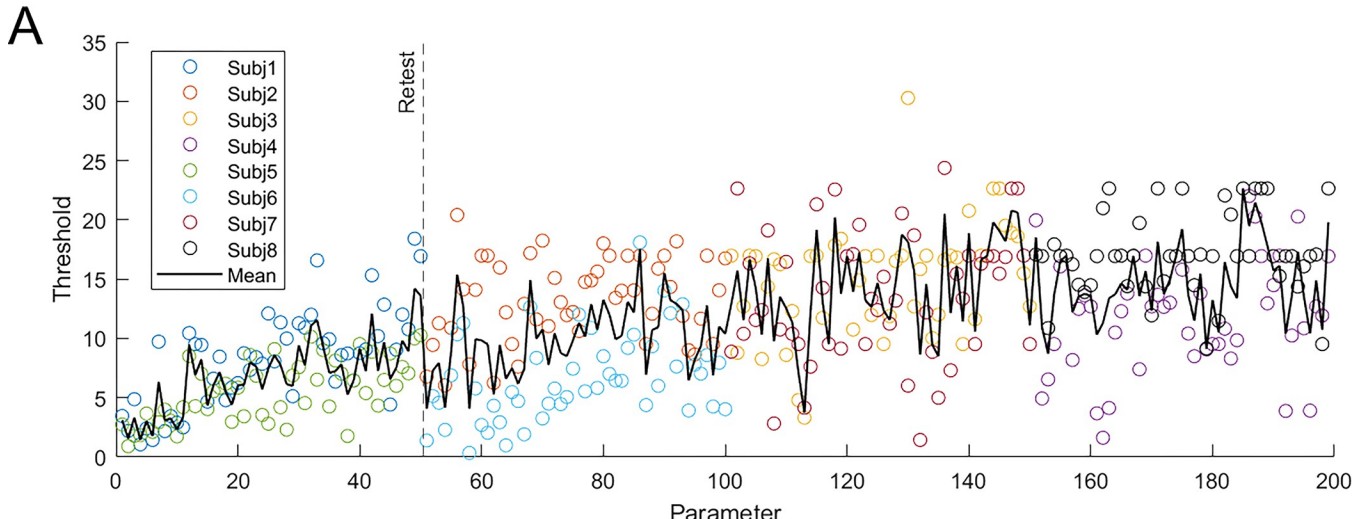

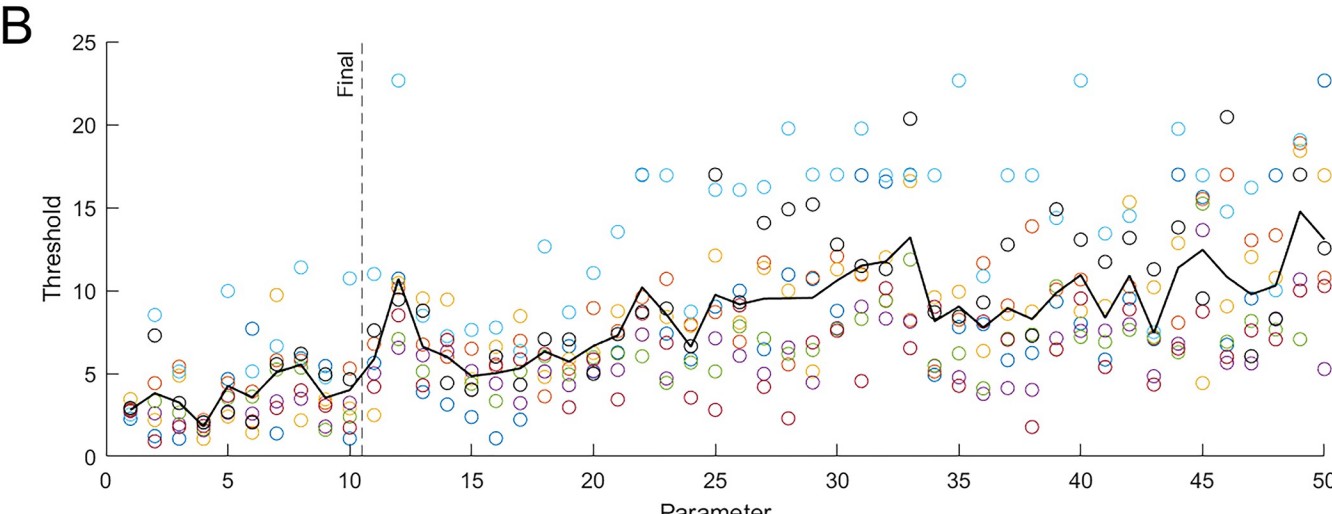

**Fig 3. Face discrimination thresholds for individual components of the Basel Face Model.** Parameter refers to the principal component of the Basel Face Model being investigated. Thresholds are defined as the difference in standard deviation units for a single component of the model at which face identities are reported as different. A) Results for all 199 structure components, each tested by 2 participants across a total of 8 participants. B) Results for structure components 1–50 for 8 participants per component. In both figures, open circles show data for individual observers by color, the mean threshold across participants is plotted as a black line.

41:50 = 10.87). We therefore investigated the first 10 components for the remainder of the studies.

**Experiment 2: Single vs combination components.** The results of Experiment 1 showed that face discrimination thresholds generally increased with the component number of the Basel Face Model. This result is consistent with the order of variance explained by the principal components of the model. Next, we explored if the thresholds for individual components are independent of one another perceptually, and to what extent changes to combinations of features affect face discrimination. We selected the first 10 components of the Basel Face Model for investigation because thresholds were lowest for these components and were less likely to saturate than components with lower sensitivity. We measured face discrimination thresholds for the first 10 components individually (single) and paired with their neighboring component

(combined). For example, single conditions measured components 1, 2, 3, etc., while combined conditions measured components 1 and 2, 2 and 3, 3 and 4, etc. When investigating combined parameters, the two components were changed to the same extent and in the same direction. We ran this study remotely over Zoom (Zoom Video Communications Inc., San Jose, CA) to test the accessibility of this method for potential clinical populations.

## Results

### Remote vs in-lab thresholds

We compared thresholds 1–10 from Experiment 1, tested in lab on a calibrated system, with the single thresholds from Experiment 2, tested remotely on an uncalibrated system. There was no significant difference thresholds in these studies (t(14) = 0.23, p = .82, d = 0.17), suggesting that this task may be successfully administered remotely. Overall, thresholds for component 4 were lowest and were significantly lower than components 5 (p = .014), 7 (p < .001), 8 (p < .001), and 10 (p = .010), but thresholds for all other components were not significantly different from one another.

### Single vs combined component thresholds

Fig 4 shows boxplots of the thresholds for single (blue) and combined (red) components. Thresholds for combined components were significantly lower than thresholds for single

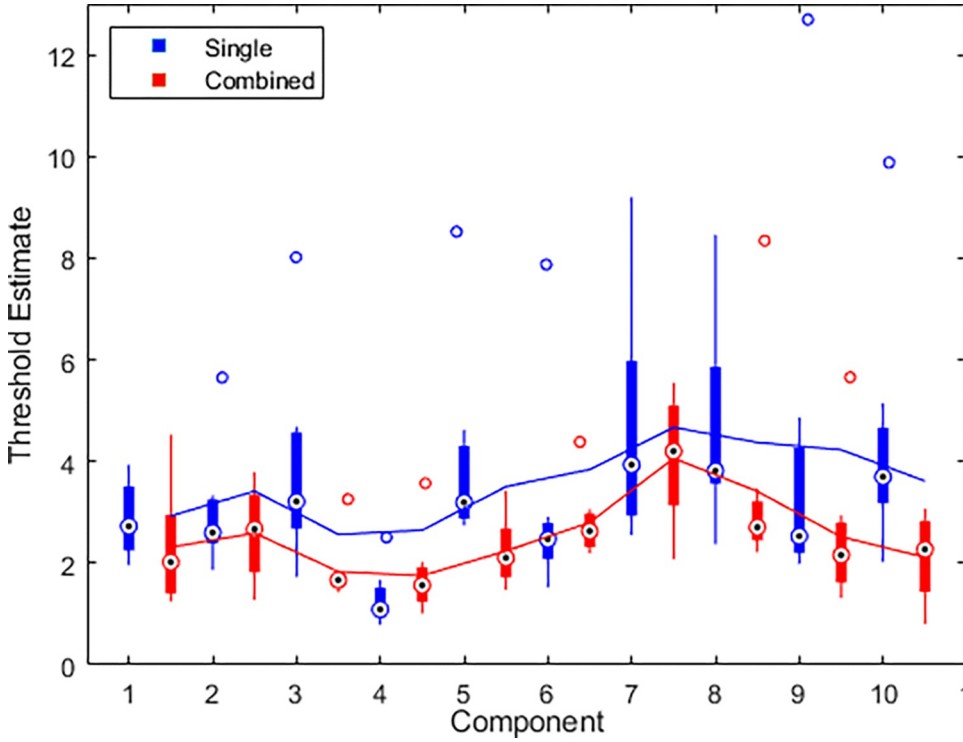

**Fig 4. Face discrimination thresholds for single and combined components of the Basel Face Model.** Blue data show results for single components, red data show results for combinations of the two components between which the bars are plotted. Circles with black dots represent medians, boxes represent IQR, whiskers represent minima and maxima, unfilled circles represent outliers, lines represent the means across components, where the blue line measures the average threshold of two single components (e.g. the mean of components 1 and 2), and the red line measures the average threshold of the combined components. 95% confidence intervals are [3.10, 4.06] for single components and [2.29, 2.84] for combined components.

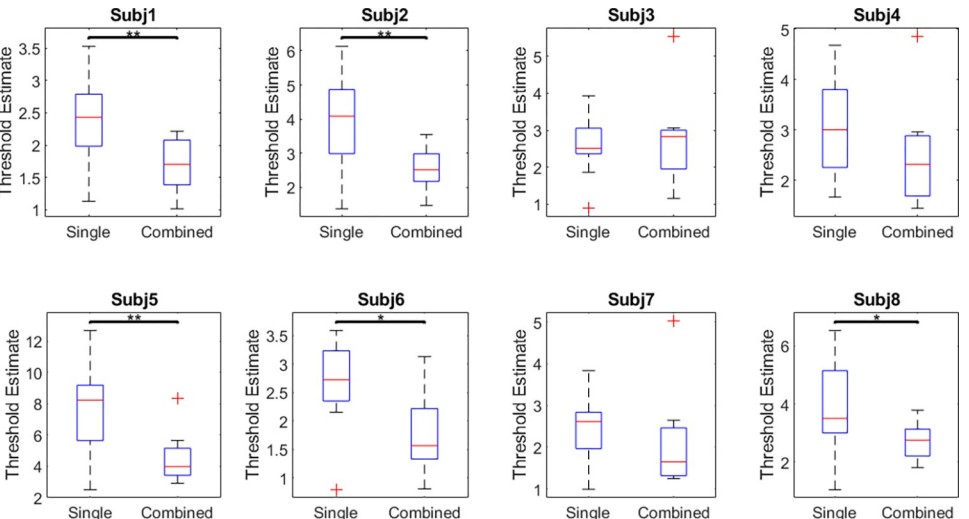

**Fig 5. Boxplots for individual participants across all 10 single and combined components.** Red crosses represent outliers, red lines represent medians, box represents IQR, whiskers represent minimums and maximums.

components by an average of about 1.396 times (t(7) = 2.90, p = .023, d = 1.03), which is not significantly different from facilitation expected from probability summation (t(7) = -0.36, p = 0.73, d = 0.13). We expect the ratio of means between single and combined thresholds to exceed $\sqrt{2}$ for the result to point towards probability summation supperadditivity. Similarly, the overall average thresholds were lower for combined components compared to single in 7 out of 8 participants, and were significantly lower for 5 of these participants (Fig 5). Each chart was completed in an average of 30.38 ± 10.00 seconds.

**Experiment 3: Rotation and single/combination.**   In Experiments 1 and 2, the null faces were pixelwise identical, which provided a potential point-wise cue to discriminate target from null face pairs. Furthermore, it is possible that participants who could free-fuse might be able to use this method to identify target and null faces. The observation in Experiment 1 that threshold depended strongly on component number suggests that participants did not use these strategies. However, to remove these potential cues, we replicated the single vs combination study of Experiment 2 with an additional manipulation of the rotation of faces. The horizontal rotation (yaw) of each face was assigned a uniform random deviate between -5° and +5°. This way, the two faces were never pixel-wise identical, and superimposition strategies could not be invoked. This also stressed the importance of faces needing to have the same identity, rather than just being identical images.

## Rotation and single/combination

Face identification thresholds in Experiment 3 with head rotation were significantly higher on average than thresholds in Experiment 2 with all faces in frontal pose (t(30) = 4.32, p < .001, d = 1.53). This is consistent with a decrease in recognition performance for non-frontal face poses [63]. When comparing single and combination components, we found a similar trend to non-rotated faces, where thresholds for combined components were approximately 1.16 times lower compared to individual components, which is not significantly different from facilitation expected from summation (t(7) = -0.40, p = .70, d = 0.14), however this did not reach significance (t(7) = 2.24, p = .061, d = 0.79). In almost all cases, threshold for combined components were lower thresholds than the average of the two components when measured

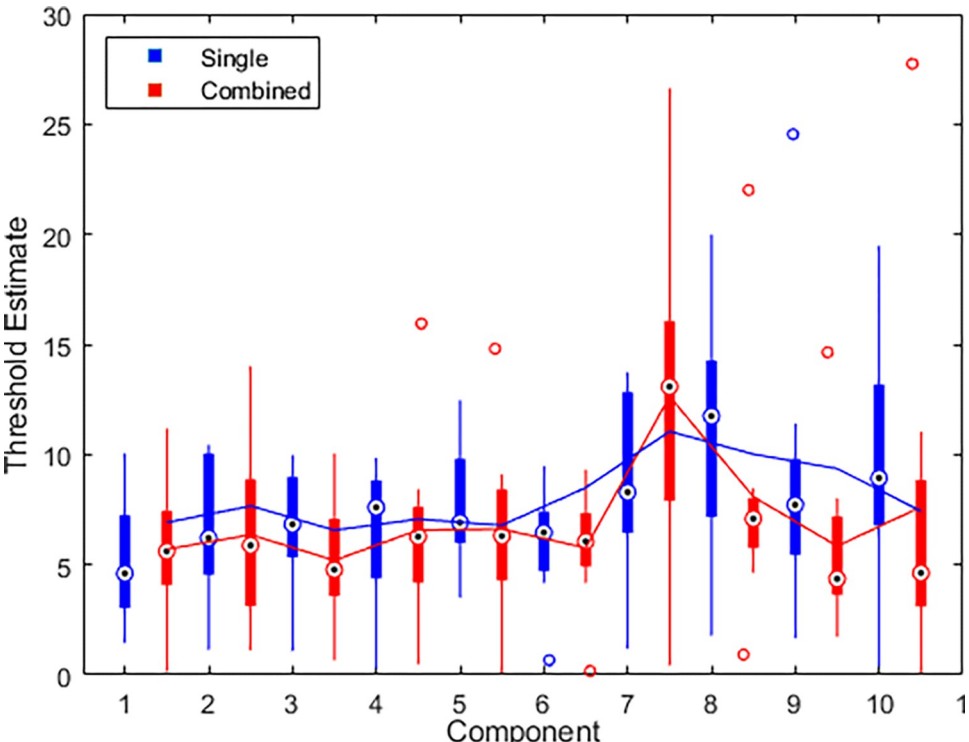

**Fig 6. Boxplots of discrimination thresholds for single components and combined components with added rotation.** Box and whisker notation as Fig 4. 95% confidence intervals are [6.90, 9.38] for single components and [5.85, 8.23] for combined components.

individually, with the exception of combined components 7 and 8 (by 1.59 std units) and combined components 10 and 1 (by 0.19 std units, Fig 6). Similarly, the overall average thresholds were lower for single components compared to combined in 6 out of 8 participants, and were significantly lower for 3 of these participants (Fig 7). Although the core task was identical,

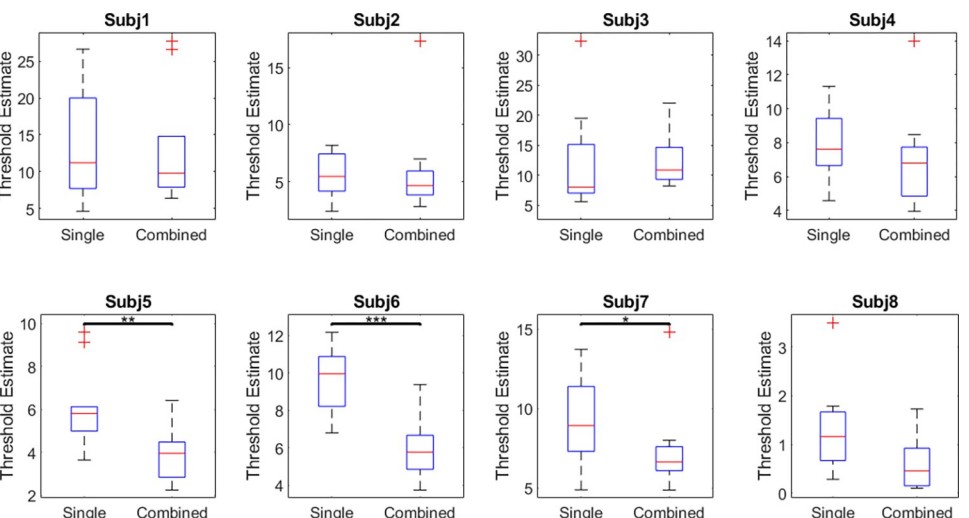

**Fig 7. Boxplots for individual participants across all 10 single and combined components with added rotation.** Box and whisker notation as Fig 5.

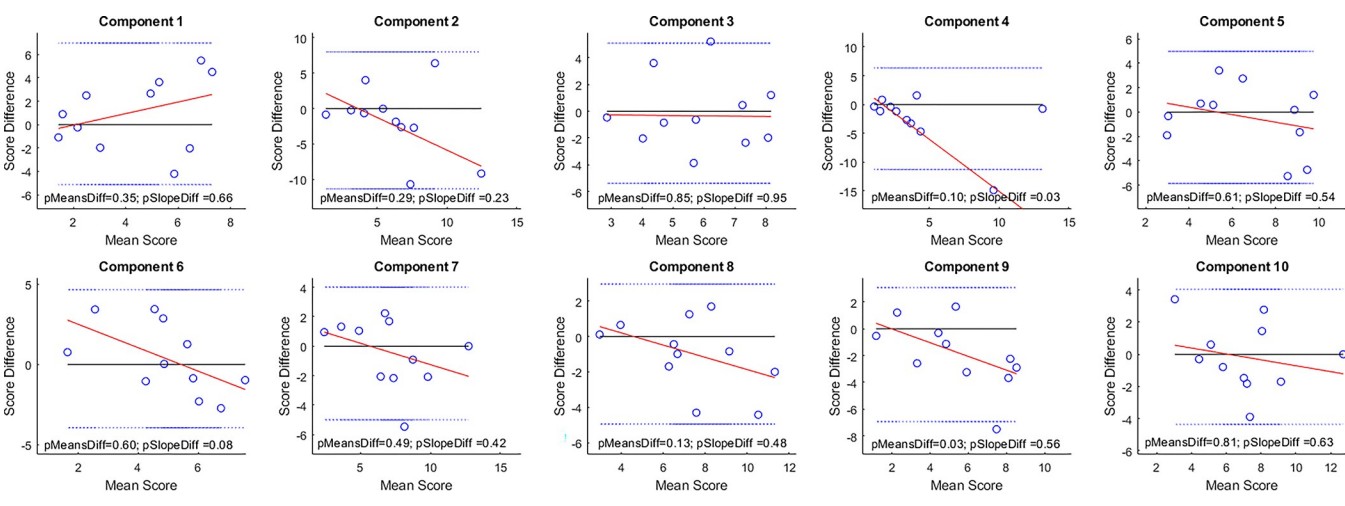

**Fig 8. Bland Altman plots for test-retest experiment across individual components.**

participants performed this rotation experiment significantly faster than the version without rotation (t(30) = 2.66, p = .012, d = 0.94). Each chart was measured in an average of 22.55 ± 6.19 seconds.

## Bland-Altman analysis

We performed an additional test-retest experiment to investigate the systematic bias and error variance heterogeneity of our task. 11 additional participants recruited from the Northeastern undergraduate population completed the 10 single components twice in the same session. We performed a Bland-Altman analysis for each of the 10 individual components. There were no significant differences between the means difference or slope difference for any of the tested components, except for the slope difference of component 4 (p = 0.03) and the means difference of component 9 (p = 0.03). The significant slope difference of component 4 seems to be mainly driven by an outlier (Fig 8).

## Discussion

The ability to recognize faces is a critical skill in daily activities and while impairments in this ability are relatively prevalent, there is an unmet need for quantitative methods for the assessment of face processing deficits. In this study, we used the Basel Face Model to generate controlled changes to morphable face structure in a self-administered psychophysical task performed either in-lab or remotely. We measured face discrimination thresholds for parametric changes along different dimensions in the face space of the Basel Face Model in healthy young participants. We investigated how face discrimination thresholds are altered when components along a single dimension were manipulated compared to when components along two dimensions in combination were manipulated, and how thresholds vary with slight horizontal head rotation. Experiment 1 showed that thresholds for the first 10 dimensions in the Basel Face Model were lowest, so our 2nd and 3rd experiments concentrated on these dimensions in single and paired combinations.

In order to explore this multi-dimensional space efficiently, we adapted the FInD paradigm for face discrimination. In this paradigm, observers viewed a series of charts each containing multiple face pairs, and judged whether each face pair was the same or different identity. The difference between faces was controlled with the adaptive FInD algorithm. The use of near-

unlimited novel face pairs avoided memory and familiarity artefacts inherent in some other approaches to assess face discrimination performance, and the use of an adaptive algorithm allowed the assessment across a range of ability levels with high precision and to track changes in performance over time. Thresholds were measured in an average of 2 minutes per component, and in future work we aim to examine which single or combined components are diagnostic for people with prosopagnosia. The study was easily converted to a remote administration with no significant loss of accuracy or efficiency. This self-administered design can be adapted for use by a variety of clinical populations, many of which have difficulties sitting still in front of a screen for extended periods of time and may struggle to visit a lab or clinic. The design can also be easily modified to accommodate, such as reducing the number of cells per trial or using a hidden cell paradigm to reduce the amount of clutter on the screen at one time. We found no differences between thresholds that were measured in-lab, where screen components and viewing distance were highly controlled, and remote testing, where these components were relatively uncontrolled. There is no time or mobility constraint, and the task is interactive and straightforward, and many breaks can be implemented into the design. This paradigm may therefore hold potential for facial processing deficit screening in a variety of populations.

Jozwik et al. (2022) measured the perceived difference between face pairs in the Basel Face Model and found good agreement between the Euclidean distance and the perceived similarity between faces. They went on to show that Euclidean distance in the Basel Face Model was as good or better at predicting perceived similarity as several image and face processing models. The results of the single component discrimination tasks in Experiments 1–3 show a strong dependence of thresholds on the Basel Face Model component, indicating that Euclidean distance along single dimensions alone does not predict perceived similarity very well. However, the results of the combined components in Experiments 2 and 3 show threshold elevation, which is in agreement with the Euclidean distance between points on different dimensions and supports the findings of Jozwik et al. (2022).

In the forward-facing single/combination study in Experiment 2, we observed significantly lower thresholds for component 4, compared to other components (Fig 4). Because component 4 mainly changes the overall width of the face (Fig 1), it is comparable to changing the face's aspect ratio. It has previously been shown that for simple shapes, aspect ratio deviations from symmetry are detected with high accuracy [64], and thus the hyperacuity for detecting aspect ratio might contribute to the detection of changes to this individual feature. This speculation is supported by the observation in Experiment 3 that sensitivity to changes in component 4 is lost when we add the head rotation, as the changes in aspect ratio are no longer as reliable once the faces are not facing in the same direction as each other.

We observed maximum sensitivity to differences in face structure when both faces in the pair were facing directly forward. When variability in the form of slight head rotation (a uniform deviate between ±5˚) was added to prevent pixel-wise comparisons, we noticed a decrease in sensitivity. Comparing thresholds for the single and combined components between Experiment 2 with forward-facing heads and Experiment 3 with random head rotation, we noticed similar trends (thresholds were around 10% lower for combined vs single component modifications), but the threshold reduction did not reach statistical significance in this study. These conditions of head tilt deviation are representative of real-world variability, as we rarely view faces at the same angle every time. The degree to which the combination of features improved discrimination was comparable in the ideal condition study (1.33) and the rotation study (1.15). Under the laws of probability summation, the degree at which the combined features lowers the threshold of discrimination does not constitute superadditivity [65], but rather follows the traditional additivity function. Through this, we can conclude that the

components we tested are independent and processed separately. When variability was added in the form of rotation, threshold values increased significantly, but the degree to which combination of features helped with discrimination remained relatively constant. The ~7% threshold increase observed in the rotation study could suggest probability summation, where the perception of these specific features are less impacted by rotational viewing.

This is supported by evidence that changes in external features (i.e. head shape) are more easily discriminated than internal features (i.e. eyes, nose, mouth) in unfamiliar faces [66]. The most salient features in the Basel Database, (the first 10 that we tested here), appear to be external and configural, affecting broadly the overall shape of the head and distribution of the face. As a result, changing a single one of these components consequently shifts multiple internal features in the process. Because changing a single component can have effects on a variety of features, it is interesting that combining components does not result in superadditivity. Under the holistic view of face processing, one might think that the more the face parts change, the more the threshold of discrimination would lower–however we find that this is not the case.

In Experiment 3, we examined the effect of random head rotations and saw a significant increase in threshold values compared with straight-ahead gaze. This is somewhat unsurprising, as the ability to discriminate differences is potentially easier with forward-facing faces because subtle changes in face parts become less apparent when the faces cannot be directly compared in a pixel-by-pixel method. The significant increase we observed is interesting because it may require participants to discriminate whether the faces belong to the same *identity*, rather than just whether the two images are *identical*. The ability to identify a single face in a variety of positions as belonging to the same identity is a natural task in daily life. We therefore recommend that this paradigm is more effective for assessing face processing performance when rotated faces are used because they require a more abstract understanding of the face identity as a whole.

Although the task was identical in Experiments 2 and 3, participants completed Experiment 3 (with added head rotation) significantly faster than Experiment 2 (where heads were aligned). A possible explanation could be that in Experiment 2, where the faces were both forward-facing, participants can use information about face identity in addition to point-wise comparisons between images and therefore used additional time to search for subtle differences between face details before making a decision. When the faces are rotated, detail-oriented scanning is no longer an effective strategy, and observers may rely exclusively on face identity without additional time on point-wise comparisons.

A limitation of this study is that the Basel Face Model suffers from some of the same problems encountered in current major prosopagnosia tests: age, race, and sex biases [36, 37, 67]. The mean face in the Basel Face Model (see Fig 1) resembles a young, white male, and as such, the majority of faces skew towards that demographic. However, where the database suffers in variety, it succeeds in specificity. The fine-tuned adjustments that can be made in morphable 3D models are unmatched to any previous tests which rely on real-world images of faces. While improvements to the model may be made by increasing racial and sexual diversity, inclusion and variety, there is still potential in the small subset which it currently holds to investigate face discrimination performance.

Overall, we demonstrate that the FInD paradigm can be generalized to a face perception task that can be translated to remote sessions. We found that healthy young observers are most sensitive to changes in the first 10 components of the Basel Face Model, and there is evidence of threshold summation when multiple components are tested in combination, consistent with an independent detection and processing of these features and with Euclidean distances in face space. We also show that face discrimination thresholds increase when slight rotation is added to the faces, which more representative of real-world variability, and may force

participants to judge differences between faces based on face identity, rather than whether the images are identical.

## Acknowledgments

The authors would like to thank Kate Storrs for help with the MATLAB interface for the Basel Face Model. The authors would like to acknowledge Prof. Dr. T. Vetter, Department of Computer Science, and the University of Basel as the source of the Basel Face Model.

## Author Contributions

**Conceptualization:** Kerri Walter, Peter Bex.

**Data curation:** Kerri Walter.

**Formal analysis:** Kerri Walter.

**Funding acquisition:** Peter Bex.

**Supervision:** Peter Bex.

**Writing – original draft:** Kerri Walter.

**Writing – review & editing:** Kerri Walter, Peter Bex.

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
