## [Decision Letter · Decision Letter 0]

6 Aug 2024

PONE-D-23-42585Rapid quantification of face discrimination performance in laboratory and remote settingsPLOS ONE

Dear Dr. Walter,

Thank you for submitting your manuscript to PLOS ONE. After careful consideration, we feel that it has merit but does not fully meet PLOS ONE’s publication criteria as it currently stands. Therefore, we invite you to submit a revised version of the manuscript that addresses the points raised during the review process.

First of all, let me apologize for the delay in getting back to you about your submission. I have now secured two expert reviews of your manuscript. As you will see below, the reviewers raised concerns about the reproducibility of your study which preclude publication of this version and which would need to be addressed in a revision of the paper. These concerns mainly revolve around the FInD procedure, which is not described in sufficient detail and for which no reference (other than a meeting abstract) is provided, and the lack of clarity in describing the use of the Basel Face Model (e.g., which version was used, handling of structure vs. texture components, procedure for combined parameter manipulation). In addition, as pointed out by Reviewer #2, some claims about the reliability of the method appear to be not sufficiently backed by the statistical analyses.

We look forward to receiving your revised manuscript.

Kind regards,

Patrick Bruns

Academic Editor

PLOS ONE

“Supported by NIH R01 EY029713”

3. We note that you have a patent relating to material pertinent to this article. Please provide an amended statement of Competing Interests to declare this patent (with details including name and number), along with any other relevant declarations relating to employment, consultancy, patents, products in development or modified products etc. Please confirm that this does not alter your adherence to all PLOS ONE policies on sharing data and materials, as detailed online in our guide for authors http://journals.plos.org/plosone/s/competing-interests by including the following statement: "This does not alter our adherence to  PLOS ONE policies on sharing data and materials.” If there are restrictions on sharing of data and/or materials, please state these. Please note that we cannot proceed with consideration of your article until this information has been declared.

4. We note that Figures 1 and 2 in your submission contain copyrighted images. All PLOS content is published under the Creative Commons Attribution License (CC BY 4.0), which means that the manuscript, images, and Supporting Information files will be freely available online, and any third party is permitted to access, download, copy, distribute, and use these materials in any way, even commercially, with proper attribution. For more information, see our copyright guidelines: http://journals.plos.org/plosone/s/licenses-and-copyright.

a. You may seek permission from the original copyright holder of Figures 1 and 2 to publish the content specifically under the CC BY 4.0 license.

Reviewers' comments:

Reviewer's Responses to Questions

**Comments to the Author**

1. Is the manuscript technically sound, and do the data support the conclusions?

Reviewer #1: Yes

Reviewer #2: No

2. Has the statistical analysis been performed appropriately and rigorously? 

Reviewer #1: Yes

Reviewer #2: No

3. Have the authors made all data underlying the findings in their manuscript fully available?

Reviewer #1: Yes

Reviewer #2: No

4. Is the manuscript presented in an intelligible fashion and written in standard English?

Reviewer #1: Yes

Reviewer #2: Yes

5. Review Comments to the Author

Reviewer #1: SUMMARY

The paper proposes and demonstrates a new efficient adaptive measure of face discrimination ability, by combining the authors' adaptive algorithm "FInD" with the smoothly-searchable PCA-based Basel Face space. The work seems soundly conducted. I have only a few suggestions for clarification:

MAJOR COMMENTS

- My one recurring point of confusion was how the separate structure vs texture components of the Basel Face Model were handled. The BFM consists of 199 dimensions capturing the 199 principal components (PCs) of variation in face mesh geometry, and another separate 199 dimensions capturing the 199 PCs of variation in RGB pixel values of the face texture. To visualise a face from the BFM, one specifies 398 numbers (the values of all structure and texture PCs), chooses a head angle and lighting, then renders the image. The structure and texture subspaces are the result of separate PCAs run on different data and don't have any relation to one another. In some places in the ms it sounds like only 199 numbers were specified, perhaps because the average texture (PCs = {0,0,0,....}) was applied to all faces, but then other parts of the ms seem to suggest that both subspaces were randomly sampled. Some sections in which to clarify this:

- line 136 "The faces are specified by 199 different parameters"

- line 139 "199 random values (...) for the structure and texture parameters"

- line 152 "All texture parameters are 0, only structure parameters are altered" (this seems clear - but comes a bit late, and seems to conflict with some of the other quoted parts)

- line 180 "same 199 random deviates (...) for the structure and texture parameters"

MINOR COMMENTS

- Following on from the previous point, it would generally be helpful to discuss the BFM more precisely in terms of PCs, rather than abstract "parameters". E.g. line 136-137 "199 different parameters each of which controls a different aspect of the face." This doesn't help a reader who isn't familiar with the BFM think about what sorts of "aspects" are being varied. It's also a bit confusing to readers who are familiar with the BFM, because there are 398 parameters per face really, or 199 *structural* parameters.

- It is interesting to see the data in Figure 3 showing face discrimination thresholds for changes in each PC separately. The fact that early PCs induce larger visual changes is clear from playing with the BFM, but I haven't seen this quantified before. I would be curious to know whether the decrease in sensitivity is matched to the decrease in *physical stimulus change* created by varying each PC. This could be quantified either in the 3D space (as Euclidean distance between the original and altered face meshes), or image space (as pixelwise difference between the two rendered images).

-

- Which version of the BFM does the work use? The issue is confusing because of the existence of the original morphable Basel Face Model in 2009, the collection of face photographs known as the Basel Face Database released in 2018, and then the new-and-improved morphable Basel Face Model version from 2019. From the figures it looks like the 2009 BFM (e.g. the face images are clearly computer-generated, so are BFM not BFD, and they have sharply cropped foreheads rather than visible hairlines pointing to 2009 rather than 2019 BFM). However, the authors reference Walker et al (2018) in several key places (e.g. Stimuli, p7), which I believe is a paper presenting the Basel Face Database, which is not a morphable model. And then the URL provided in the code repository to download the BFM points to the 2019 version (Gerig et al 2019 - which isn't cited). Please clarify throughout. It would also be good to include a brief discussion note on the distinctions between the 2009 and 2019 BFM version - e.g. could using the 2019 version help improve the diversity of the faces?

- Are other researchers intended to use this method? The main goal of the paper seems to be to promote this task as an efficient method for measuring face discrimination ability. Code for the task appears online, suggesting that any researcher able to use PsychToolbox is free to run their own version of this experiment. However, the adaptive FInD algorithm underlying it is protected by a patent belonging to one of the authors. It would be good to clarify the status/intention - are researchers encouraged to use the freely available code, and meanwhile the algorithm may be built into a more user-friendly but paid app for clinicians or patients to use?

- line 57 "The (BFM) face space is isometric" - what is the evidence for this? In fact it seems clearly untrue as the data and examples in this paper show; moving 1 SD along an early principal component creates a much more visible change than moving 1 SD along a late component. It also creates a larger physical change in the stimulus (as mentioned above).

- line 168 The power analysis mentioned here specifies a very large expected effect size (1 SD) - what is this the difference between, and in what statistical analysis? Was this effect size estimate arrived at prior to the experiment, and if so how?

- line 316 It's not clear to me why this task would be better suited, in principle, to track changes in face perception, compared to other tasks?

TYPOS etc

- line 26 "Thresholds measured remotely WERE not..."

- line 67 "there of" >> "thereof"

- line 96 what does the "participantive impact" of prosopagnosia mean?

- line 115 "it's" >> "its"

- line 117 "stereoacuity (...), AND motion"

- line 166 "Additionally" - not clear why these participant demographics are reported separately

Reviewer #2: This study explores human face discrimination thresholds using a novel approach. It employs the Basel Face Model to generate synthetic, parametrized face stimuli that are adaptively adjusted to estimate the discrimination thresholds specific to each BFM parameter. The main empirical finding is an inhomogeneity in how different BFM parameters relate to human face discrimination. This result advances the state of the art beyond what was previously described by Jozwik et al., 2022.

Unfortunately, this work relies on an unpublished and not fully disclosed adaptive psychophysical procedure, “FInD”. In my opinion, this omission raises significant concerns that must be addressed before the manuscript can be considered for publication. Additionally, I believe that there are several other major issues that must be corrected before the paper can be published, as described below.

Major

1.The FInD procedure, which is central to the current work, is not described in sufficient detail. The reference for the procedure is a VSS meeting abstract rather than a peer-reviewed paper. This issue becomes evident only after following the DOI link, as the reference initially appears to be a Journal of Vision paper.

Without a formal description and validation of the FInD procedure, many questions remain unanswered. For example: a. What Bayesian model is used? b. What assumptions does it make? c. How were the priors chosen? d. How is the efficiency metric calculated? e. What is the justification for Eq. 1? f. How is the potential interaction between cells taken (or not) into account? (The observer can quickly infer that there’s only a single positive cell and use this regularity in their decision-making). g. How reliable is the method compared to alternatives?

Without these details, fully evaluating this work is difficult. I suggest the authors either significantly revise the paper to include a complete description and validation of the FInD procedure or cite a detailed (preferably peer-reviewed) report on this method.

2. The stimulus manipulation in Experiment 2 is not described in sufficient detail. What does it mean for the parameters to be manipulated in a combined fashion? Were both parameters changed to the same extent and in the same direction (i.e., both increased or both decreased)?

3. While the paper makes claims about the reliability of the method (e.g., “the use of an adaptive algorithm allowed the assessment across a range of ability levels with high precision”), it does not establish these claims. Failing to find significant differences between replications does not adequately demonstrate reliability, as null hypothesis testing does not control the type II error rate. Furthermore, for a potential diagnostic tool, the reliability of interest pertains to individual-level estimates rather than average estimates pooled from an entire group of subjects. I believe that to establish claims about reliability, measures such as test-retest reliability must be formally reported and compared to alternative approaches.

4. Judging from the VSS abstract, the model appears to be Bayesian, allowing for the calculation of a credible interval for each parameter estimate. Such intervals are not indicated, nor is any other measure of individual-level estimation error. Claiming that the estimation is “precise” without quantifying estimation error is problematic.

Minor:

- Lines 253 and 256 seem inconsistent: “Thresholds for combined parameters were significantly lower than thresholds for single parameters”, versus “the overall average thresholds were lower for single parameters compared to combined in 7 out of 8 participants”.

- Figure 3 displays horizontal stripes of measurements of equal threshold (around y=16). Is this an estimation artifact?

- It would be useful to spell out mathematically the predictions of probability summation in the current context.

- Presenting figures at the end of the manuscript, disconnected from their captions, makes reviewing very cumbersome.

6. PLOS authors have the option to publish the peer review history of their article (what does this mean?). If published, this will include your full peer review and any attached files.

Reviewer #1: No

Reviewer #2: No

---

## [Author Response · Author response to Decision Letter 0]

26 Sep 2024

Reviewer #1: 

The paper proposes and demonstrates a new efficient adaptive measure of face discrimination ability, by combining the authors' adaptive algorithm "FInD" with the smoothly-searchable PCA-based Basel Face space. The work seems soundly conducted. I have only a few suggestions for clarification: 

MAJOR COMMENTS 

- My one recurring point of confusion was how the separate structure vs texture components of the Basel Face Model were handled. The BFM consists of 199 dimensions capturing the 199 principal components (PCs) of variation in face mesh geometry, and another separate 199 dimensions capturing the 199 PCs of variation in RGB pixel values of the face texture. To visualise a face from the BFM, one specifies 398 numbers (the values of all structure and texture PCs), chooses a head angle and lighting, then renders the image. The structure and texture subspaces are the result of separate PCAs run on different data and don't have any relation to one another. In some places in the ms it sounds like only 199 numbers were specified, perhaps because the average texture (PCs = {0,0,0,....}) was applied to all faces, but then other parts of the ms seem to suggest that both subspaces were randomly sampled. Some sections in which to clarify this: 

- line 136 "The faces are specified by 199 different parameters" 

- line 139 "199 random values (...) for the structure and texture parameters" 

- line 152 "All texture parameters are 0, only structure parameters are altered" (this seems clear - but comes a bit late, and seems to conflict with some of the other quoted parts) 

- line 180 "same 199 random deviates (...) for the structure and texture parameters" 

Thank you, we have clarified the description in the revised manuscript in the section “Stimuli”. 

In summary, we create unique random values for the 199 structural and textural Basel Face Model principal components for the face pair in each cell. The face pair within a cell share the same set of random values for 198 structural and 199 textural components; but component values differ across cells. Within each face pair, only a single target structural component differs, and all textural components remain constant. Therefore, the faces within a cell are similar (by a near-threshold distance), but faces between cells are very different.. 

MINOR COMMENTS 

- Following on from the previous point, it would generally be helpful to discuss the BFM more precisely in terms of PCs, rather than abstract "parameters". E.g. line 136-137 "199 different parameters each of which controls a different aspect of the face." This doesn't help a reader who isn't familiar with the BFM think about what sorts of "aspects" are being varied. It's also a bit confusing to readers who are familiar with the BFM, because there are 398 parameters per face really, or 199 *structural* parameters. 

Thank you, we have added text to explain that the BFM is a PC model with separate structural and textural components, and we use the term ‘component’ to refer to the parameters throughout the revised manuscript. 

- It is interesting to see the data in Figure 3 showing face discrimination thresholds for changes in each PC separately. The fact that early PCs induce larger visual changes is clear from playing with the BFM, but I haven't seen this quantified before. I would be curious to know whether the decrease in sensitivity is matched to the decrease in *physical stimulus change* created by varying each PC. This could be quantified either in the 3D space (as Euclidean distance between the original and altered face meshes), or image space (as pixelwise difference between the two rendered images). 

We agree that this is an interesting comparison, and we understand that principal components are already ordered according to the variance in the data. One of the goals of our paper was to compare the perceptual sensitivity with image variance captured in the BFM components. 

- Which version of the BFM does the work use? The issue is confusing because of the existence of the original morphable Basel Face Model in 2009, the collection of face photographs known as the Basel Face Database released in 2018, and then the new-and-improved morphable Basel Face Model version from 2019. From the figures it looks like the 2009 BFM (e.g. the face images are clearly computer-generated, so are BFM not BFD, and they have sharply cropped foreheads rather than visible hairlines pointing to 2009 rather than 2019 BFM). However, the authors reference Walker et al (2018) in several key places (e.g. Stimuli, p7), which I believe is a paper presenting the Basel Face Database, which is not a morphable model. And then the URL provided in the code repository to download the BFM points to the 2019 version (Gerig et al 2019 - which isn't cited). Please clarify throughout. It would also be good to include a brief discussion note on the distinctions between the 2009 and 2019 BFM version - e.g. could using the 2019 version help improve the diversity of the faces? 

We utilize the 2019 version of the BFM, this has been clarified in the text and the Gerig paper has been cited. 

- Are other researchers intended to use this method? The main goal of the paper seems to be to promote this task as an efficient method for measuring face discrimination ability. Code for the task appears online, suggesting that any researcher able to use PsychToolbox is free to run their own version of this experiment. However, the adaptive FInD algorithm underlying it is protected by a patent belonging to one of the authors. It would be good to clarify the status/intention - are researchers encouraged to use the freely available code, and meanwhile the algorithm may be built into a more user-friendly but paid app for clinicians or patients to use? 

We have clarified (Line 110) that the patent for FInD protects commercialization of the method, but researchers are free and encouraged to use the method for non-profit purposes. If the method can successfully quantify face processing differences (e.g. the presence or remediation of prosopagnosia), the longer-term goal would be to develop and distribute a user-friendly test. 

- line 57 "The (BFM) face space is isometric" - what is the evidence for this? In fact it seems clearly untrue as the data and examples in this paper show; moving 1 SD along an early principal component creates a much more visible change than moving 1 SD along a late component. It also creates a larger physical change in the stimulus (as mentioned above). 

The space is isometric in that equal Euclidean distance between face pairs renders them the same amount of “different” (see https://doi.org/10.1073/pnas.2115047119). However, we agree with the reviewer that moving along early PCs creates larger visual variation than moving along late PCs. 

We have amended this statement (Line 55), which was originally cited from Blanz V & Vetter T (1999) & Paysan P et. al. (2009). 

- line 168 The power analysis mentioned here specifies a very large expected effect size (1 SD) - what is this the difference between, and in what statistical analysis? Was this effect size estimate arrived at prior to the experiment, and if so how? 

As this was an exploratory study, we ran a small pilot study that demonstrated robust effects. We have added some clarification in the main text. 

- line 316 It's not clear to me why this task would be better suited, in principle, to track changes in face perception, compared to other tasks? 

There are several features of this method that afford benefits over other tasks. Primarily the faces for each trial are unique, so the test can be repeated without confounds of stimulus learning or familiarity. Furthermore, the test avoids memory effects and recognition of known individuals. Ultimately, we propose that this task and could be it is a simple, rapid, effective could complement current methods that assess face perception. 

TYPOS etc 

- line 26 "Thresholds measured remotely WERE not..." 

Thank you, corrected. 

- line 67 "there of" >> "thereof" 

Thank you, corrected. 

- line 96 what does the "participantive impact" of prosopagnosia mean? 

This word has been clarified to “participant’s perceived”. 

- line 115 "it's" >> "its" 

Thank you, corrected. 

- line 117 "stereoacuity (...), AND motion" 

Thank you, corrected. 

- line 166 "Additionally" - not clear why these participant demographics are reported separately 

Following NIH guidelines, the questionnaire contained a separate question for “Latino or Hispanic Origin”, distinct from “Race”, because individuals of Hispanic origin may be of any race. 

Reviewer #2: 

This study explores human face discrimination thresholds using a novel approach. It employs the Basel Face Model to generate synthetic, parametrized face stimuli that are adaptively adjusted to estimate the discrimination thresholds specific to each BFM parameter. The main empirical finding is an inhomogeneity in how different BFM parameters relate to human face discrimination. This result advances the state of the art beyond what was previously described by Jozwik et al., 2022. 

Unfortunately, this work relies on an unpublished and not fully disclosed adaptive psychophysical procedure, “FInD”. In my opinion, this omission raises significant concerns that must be addressed before the manuscript can be considered for publication. Additionally, I believe that there are several other major issues that must be corrected before the paper can be published, as described below. 

Major 

1.The FInD procedure, which is central to the current work, is not described in sufficient detail. The reference for the procedure is a VSS meeting abstract rather than a peer-reviewed paper. This issue becomes evident only after following the DOI link, as the reference initially appears to be a Journal of Vision paper. 

Without a formal description and validation of the FInD procedure, many questions remain unanswered. For example: a. What Bayesian model is used? b. What assumptions does it make? c. How were the priors chosen? d. How is the efficiency metric calculated? e. What is the justification for Eq. 1? f. How is the potential interaction between cells taken (or not) into account? (The observer can quickly infer that there’s only a single positive cell and use this regularity in their decision-making). g. How reliable is the method compared to alternatives? 

There are now 2 peer-reviewed publications that use the FInD method and both are now cited in the revised manuscript. 

a. The method doesn’t involve any Bayesian models, 

b. The method only assumes d’ increases with signal intensity and that a decision boundary increases with d’ (see d. below) 

c. There are no priors, 

d. Efficiency is not calculated, but we have added repeatability data in a Bland-Altman analysis 

e. We have added details to description of Equation 1to clarify that it is a saturating function of d’ as a function of stimulus intensity, based on computational (Foley & Legge, 1991) and physiological (Albrecht and Geisler, 1991) studies; followed by a decision stage (Gu and Green, 1994; Klein, 2004) 

f. There are no assumptions about interactions between cells. The number of signal cells each chart is random, so the participant cannot infer anything about the probability that a given cell is a signal or a null cell based on their responses to other cells. 

g. Based on our other studies, the FInD method is around TODO times faster than standard 2AFC psychophysics, depending on trial duration and the use of spatial or temporal alternatives. 

Without these details, fully evaluating this work is difficult. I suggest the authors either significantly revise the paper to include a complete description and validation of the FInD procedure or cite a detailed (preferably peer-reviewed) report on this method. 

Two additional peer-reviewed papers detailing the FInD method have been cited. 

2. The stimulus manipulation in Experiment 2 is not described in sufficient detail. What does it mean for the parameters to be manipulated in a combined fashion? Were both parameters changed to the same extent and in the same direction (i.e., both increased or both decreased)? 

When investigating combined parameters, two PCs were changed an equal amount. Specifically, target PC1 and target PC2 were both altered by the determined threshold deviation. Thus probability summation predicts an increase in sensitivity (decrease in thresholds) by a factor of √2. 

This has been clarified in the text under “Experiment 2”. 

3. While the paper makes claims about the reliability of the method (e.g., “the use of an adaptive algorithm allowed the assessment across a range of ability levels with high precision”), it does not establish these claims. Failing to find significant differences between replications does not adequately demonstrate reliability, as null hypothesis testing does not control the type II error rate. Furthermore, for a potential diagnostic tool, the reliability of interest pertains to individual-level estimates rather than average estimates pooled from an entire group of subjects. I believe that to establish claims about reliability, measures such as test-retest reliability must be formally reported and compared to alternative approaches. 

We have performed an additional test-retest experiment and included a description and results at the end of the Results section. 

4. Judging from the VSS abstract, the model appears to be Bayesian, allowing for the calculation of a credible interval for each parameter estimate. Such intervals are not indicated, nor is any other measure of individual-level estimation error. Claiming that the estimation is “precise” without quantifying estimation error is problematic. 

The raw data from all charts are fit with Equation 1 after each chart using Matlab’s fit() function and we estimate confidence intervals for each participant. To avoid clutter in the figures, we show box and whisker plots for each figure and have added the range of 95% confidence intervals in the figure legend. 

Minor: 

- Lines 253 and 256 seem inconsistent: “Thresholds for combined parameters were significantly lower than thresholds for single parameters”, versus “the overall average thresholds were lower for single parameters compared to combined in 7 out of 8 participants”. 

Thank you for noticing this semantic error, the secondary sentence has been amended such that single and combined are flipped. 

- Figure 3 displays horizontal stripes of measurements of equal threshold (around y=16). Is this an estimation artifact? 

Due to the PC nature of the Basel Face Model’s structure, the later PCs do not invoke as much change as early PCs. We start to see this banding in the later PCs (past 100), likely because this is the approximate mean for thresholds at these higher components. 

- It would be useful to spell out mathematically the predictions of probability summation in the current context. 

We expect the ratio of means between single and combined thresholds to exceed 1.4 for the result to point towards probability summation supperadditivity. We have clarified this in the text under “Single vs combined PC thresholds”. 

- Presenting figures at the end of the manuscript, disconnected from their captions, makes reviewing very cumbersome. 

Apologies, we are following the formatting guidelines of PLOS.

---

## [Decision Letter · Decision Letter 1]

22 Oct 2024

PONE-D-23-42585R1Rapid quantification of face discrimination performance in laboratory and remote settingsPLOS ONE

Dear Dr. Walter,

Thank you for submitting your manuscript to PLOS ONE. After careful consideration, we feel that it has merit but does not fully meet PLOS ONE’s publication criteria as it currently stands. Therefore, we invite you to submit a revised version of the manuscript that addresses the points raised during the review process.

As you will see below, while some of the reviewers' previous concerns have been clarified in the revision, there are still remaining concerns about the reproducibility of the FInD procedure as well as the test-retest reliability of the method. Please try to thoroughly address these concerns with your next revision.

We look forward to receiving your revised manuscript.

Kind regards,

Patrick Bruns

Academic Editor

PLOS ONE

Reviewers' comments:

Reviewer's Responses to Questions

**Comments to the Author**

1. If the authors have adequately addressed your comments raised in a previous round of review and you feel that this manuscript is now acceptable for publication, you may indicate that here to bypass the “Comments to the Author” section, enter your conflict of interest statement in the “Confidential to Editor” section, and submit your "Accept" recommendation.

Reviewer #1: All comments have been addressed

Reviewer #2: (No Response)

2. Is the manuscript technically sound, and do the data support the conclusions?

Reviewer #1: Yes

Reviewer #2: Partly

3. Has the statistical analysis been performed appropriately and rigorously? 

Reviewer #1: Yes

Reviewer #2: No

4. Have the authors made all data underlying the findings in their manuscript fully available?

Reviewer #1: Yes

Reviewer #2: Yes

5. Is the manuscript presented in an intelligible fashion and written in standard English?

Reviewer #1: Yes

Reviewer #2: Yes

6. Review Comments to the Author

Reviewer #1: I thank the authors for their edits to the paper, which carefully address the few points of confusion I had about the original version. I have no further comments, and think the data will be interesting to other researchers working with the BFM!

Reviewer #2: 1. I appreciate the additional details on the FInD procedure. Unfortunately, the text remains insufficient for fully reproducing the experiment. Since the other papers cited are also applications of FInD rather than methods papers fully specifying the procedure, the method description in the current paper must be complete. For example, there is no mathematical specification of how the data is fitted. Is it maximum likelihood? Least squares fitting? Based on the defaults of MATLAB's fit() function (a function mentioned in the rebuttal but not in the paper), it appears to be the latter, but these details should be stated in the paper. Also, the source code shows that observation weighting (represented as 1./errEst) was applied, but this detail is not mentioned in the paper. Complete specification of the algorithm is especially important since the provided code is in MATLAB, a platform many researchers have transitioned away from in favor of Python or R. To ensure long-term reproducibility and accessibility to a broader scientific audience, the testing methods should be fully specified in the paper.

2. Bland-Altman analysis focuses on systematic bias and is typically used to assess the agreement between two measurement tools on average. Its application here seems misguided since the primary error component of interest in test-retest analysis is variance, not bias. The current analysis does not measure variance; unbiased yet completely random measurements would also fail to show significant differences. A more appropriate measure of reliability would be the linear correlation between the first and repeated measurements, calculated across participants, separately for each component. For the correlation coefficient to be meaningful, it should be compared to equivalent estimates obtained using alternative methods (e.g., 2AFC).

3. Unless the authors establish reliability and efficiency empirically (e.g., as outlined in the previous point), I suggest the last sentence of the abstract, “Our results highlight the effectiveness and efficiency of this paradigm to measure face discrimination ability in-lab or remotely,” should be removed, as it is not directly supported by the analyses presented in the current study.

7. PLOS authors have the option to publish the peer review history of their article (what does this mean?). If published, this will include your full peer review and any attached files.

Reviewer #1: No

Reviewer #2: No

---

## [Author Response · Author response to Decision Letter 1]

29 Oct 2024

Reviewer #2:

1. I appreciate the additional details on the FInD procedure. Unfortunately, the text remains insufficient for fully reproducing the experiment. Since the other papers cited are also applications of FInD rather than methods papers fully specifying the procedure, the method description in the current paper must be complete. For example, there is no mathematical specification of how the data is fitted. Is it maximum likelihood? Least squares fitting? Based on the defaults of MATLAB's fit() function (a function mentioned in the rebuttal but not in the paper), it appears to be the latter, but these details should be stated in the paper. Also, the source code shows that observation weighting (represented as 1./errEst) was applied, but this detail is not mentioned in the paper. Complete specification of the algorithm is especially important since the provided code is in MATLAB, a platform many researchers have transitioned away from in favor of Python or R. To ensure long-term reproducibility and accessibility to a broader scientific audience, the testing methods should be fully specified in the paper.

We utilize the default fit() function with the default settings, which is the non-linear least squares method. The Matlab function fittype() is given Equation 1 to create a specialized fit model that is then used in fit(). These additional details have been added to the methods section. 

2. Bland-Altman analysis focuses on systematic bias and is typically used to assess the agreement between two measurement tools on average. Its application here seems misguided since the primary error component of interest in test-retest analysis is variance, not bias. The current analysis does not measure variance; unbiased yet completely random measurements would also fail to show significant differences. A more appropriate measure of reliability would be the linear correlation between the first and repeated measurements, calculated across participants, separately for each component. For the correlation coefficient to be meaningful, it should be compared to equivalent estimates obtained using alternative methods (e.g., 2AFC).

We argue that Bland-Altman does test for systematic variance and is represented in the plots as horizontal blue lines. Additionally, we argue that correlation coefficients for test-retest are unreliable and overestimated when working with small sample sizes (<15), and Bland-Altman should be used instead. Some reasons why the Bland-Altman method is often preferred over the correlation coefficient for test-retest repeatability assessment include:

• Measure of Agreement vs. Association:

o The correlation coefficient measures the strength of a linear relationship between two variables, but it does not assess agreement. High correlation can occur even if there is a systematic difference between measurements.

o The Bland-Altman plot, on the other hand, directly assesses agreement by plotting the differences between two measurements against their averages. This method highlights any systematic bias and the limits of agreement, providing a clearer picture of repeatability.

• Detection of Bias:

o The correlation coefficient does not reveal any bias between measurements. Two sets of measurements can have a high correlation even if one consistently overestimates or underestimates the other.

o The Bland-Altman plot can detect both fixed and proportional bias, making it easier to identify and correct systematic errors.

• Handling of Measurement Error:

o The correlation coefficient can be misleading if the measurement error is large relative to the variability between subjects.

o The Bland-Altman plot accounts for measurement error by showing the spread of differences, which helps in understanding the precision and repeatability of the measurements.

Overall, while the correlation coefficient is useful for assessing the strength of a relationship, we believe the Bland-Altman plot provides a more comprehensive assessment of agreement and repeatability, making it a better choice for this test-retest reliability study. For reference, please see https://support.sas.com/resources/papers/proceedings18/1815-2018.pdf

Regardless, we have performed a correlation analysis for each of the 10 individual components as per the reviewer’s request. Specifically, we found the linear correlation between the first and repeated measurements, across participants, separated across components. We found that the components 7 through 10 had significant correlations.

3. Unless the authors establish reliability and efficiency empirically (e.g., as outlined in the previous point), I suggest the last sentence of the abstract, “Our results highlight the effectiveness and efficiency of this paradigm to measure face discrimination ability in-lab or remotely,” should be removed, as it is not directly supported by the analyses presented in the current study.

As outlined above, we believe that the Bland-Altman analysis included in the manuscript establishes reliability in our paradigm.

---

## [Decision Letter · Decision Letter 2]

8 Nov 2024

PONE-D-23-42585R2Rapid quantification of face discrimination performance in laboratory and remote settingsPLOS ONE

Dear Dr. Walter,

Thank you for submitting your manuscript to PLOS ONE. After careful consideration, we feel that it has merit but does not fully meet PLOS ONE’s publication criteria as it currently stands. Therefore, we invite you to submit a revised version of the manuscript that addresses the points raised during the review process.

As you will see below, Reviewer #2 still has substantive concerns about the claims regarding the test-retest reliability of the method which are presented in the paper. For the paper to become acceptable for publication, these claims must either be sufficiently supported by the data or must be removed. As we usually try to avoid protracted review processes with multiple rounds of revisions, I would like to reach a final decision on this manuscript after the next round. Thus, please try your utmost best to address the remaining concerns thoroughly with your next revision.

We look forward to receiving your revised manuscript.

Kind regards,

Patrick Bruns

Academic Editor

PLOS ONE

Reviewers' comments:

Reviewer's Responses to Questions

**Comments to the Author**

1. If the authors have adequately addressed your comments raised in a previous round of review and you feel that this manuscript is now acceptable for publication, you may indicate that here to bypass the “Comments to the Author” section, enter your conflict of interest statement in the “Confidential to Editor” section, and submit your "Accept" recommendation.

Reviewer #2: (No Response)

2. Is the manuscript technically sound, and do the data support the conclusions?

Reviewer #2: Partly

3. Has the statistical analysis been performed appropriately and rigorously? 

Reviewer #2: No

4. Have the authors made all data underlying the findings in their manuscript fully available?

Reviewer #2: Yes

5. Is the manuscript presented in an intelligible fashion and written in standard English?

Reviewer #2: Yes

6. Review Comments to the Author

Reviewer #2: I appreciate the authors' detailed rebuttal. I have reviewed the reference provided by the authors and consulted additional sources.

I agree that the vertical spread in the Bland-Altman plot, quantified by the standard deviation of the differences, reflects the extent of measurement error. To determine whether the standard deviation is sufficiently small, it must be compared with a meaningful quantity: the magnitude of clinically significant differences, the error of an alternative measurement tool, or inter-subject variability. Since none of these comparisons were conducted, the Bland-Altman plots in Figure 8 cannot be used to assess test-retest reliability. Importantly, even a measure with zero test-retest reliability can still produce a benign Bland-Altman plot, showing no significant mean or slope differences and no outliers beyond two standard deviations. Therefore, the test-retest reliability subsection (lines 311–318) is misleading, as it does not evaluate reliability at all; instead, the tests reported assess systematic bias (difference mean) or error variance heterogeneity (difference slope).

I agree with the authors that linear correlation is positively biased for small samples and quantifies only reliability rather than agreement. This renders linear correlation a necessary but insufficient measure for reliability. As noted in the rebuttal, the test-retest linear correlation is not significantly different from chance for six out of ten components. Without test-retest reliability, we cannot have test-retest agreement.

Unlike the linear correlation, Lin's Concordance Correlation Coefficient (CCC) captures both variance and bias. I applied Lin’s CCC to component 1, using data from the rebuttal’s scatter plot. This yielded a value of approximately 0.02, indicating extremely poor agreement. This result heightens the concern that reliable measurement of individual, participant-specific thresholds has not been established.

I believe that the manuscript provides some interesting observations on the thresholds on average, across observers. Currently, it does not support its claims about precision, efficiency, or reliability of the measurement method itself at the individual level. The proposed method may be superior to traditional alternatives; however, the small sample size and low inter-subject variability among the 11 neurotypical participants likely limit our ability to confirm this. The individual level is important, as medical/neuropsychological diagnosis is explicitly mentioned as the general motivation for the proposed method (lines 78–88).

In my opinion, there are two potential directions for proceeding:

1. Rigorously study the reliability of the proposed method and its agreement with alternative tools by collecting a substantially larger participant sample (potentially including clinically relevant variability), analyzing test-retest agreement and between-method agreement using Lin’s CCC (in addition to Bland-Altman plots), and comparing both bias and variance with those of alternative measures (e.g., 2AFC).

2. Rewrite the paper to focus on the trends shared across subjects. This would require removing all claims of efficacy and reliability, as well as the diagnostic framing and motivation. In that case, the title should be changed as well.

7. PLOS authors have the option to publish the peer review history of their article (what does this mean?). If published, this will include your full peer review and any attached files.

Reviewer #2: No

---

## [Author Response · Author response to Decision Letter 2]

13 Nov 2024

"In my opinion, there are two potential directions for proceeding:

1. Rigorously study the reliability of the proposed method and its agreement with alternative tools by collecting a substantially larger participant sample (potentially including clinically relevant variability), analyzing test-retest agreement and between-method agreement using Lin’s CCC (in addition to Bland-Altman plots), and comparing both bias and variance with those of alternative measures (e.g., 2AFC)."

In other published works (cited below) we show that the FInD method is comparable to traditional 2AFC tasks – but because no face discrimination test like this currently exists, we cannot directly compare this face discrimination task to a similar 2AFC task. We agree that to confirm agreement with similar tests requires future work. Because of this we have rephrased the manuscript to focus on the novelty of the paradigm and have removed claims of efficacy and reliability.

doi:10.1167/jov.22.14.4351

doi:10.1167/jov.21.9.2817

doi:10.1371/journal.pone.0305036

https://iovs.arvojournals.org/article.aspx?articleid=2774587

"2. Rewrite the paper to focus on the trends shared across subjects. This would require removing all claims of efficacy and reliability, as well as the diagnostic framing and motivation. In that case, the title should be changed as well."

We have removed claims of efficacy and reliability throughout the manuscript and have edited the title to focus on the novelty of this paradigm: “A novel quantitative assessment of face discrimination ability”.

---

## [Decision Letter · Decision Letter 3]

28 Nov 2024

PONE-D-23-42585R3A novel quantitative assessment of face discrimination abilityPLOS ONE

Dear Dr. Walter,

Thank you for submitting your manuscript to PLOS ONE. After careful consideration, we feel that it has merit but does not fully meet PLOS ONE’s publication criteria as it currently stands. Therefore, we invite you to submit a revised version of the manuscript that addresses the points raised during the review process.

Please address the reviewer's remaining concern regarding the title of the manuscript. 

We look forward to receiving your revised manuscript.

Kind regards,

Patrick Bruns

Academic Editor

PLOS ONE

Journal Requirements:

Reviewers' comments:

Reviewer's Responses to Questions

**Comments to the Author**

1. If the authors have adequately addressed your comments raised in a previous round of review and you feel that this manuscript is now acceptable for publication, you may indicate that here to bypass the “Comments to the Author” section, enter your conflict of interest statement in the “Confidential to Editor” section, and submit your "Accept" recommendation.

Reviewer #2: (No Response)

2. Is the manuscript technically sound, and do the data support the conclusions?

Reviewer #2: Yes

3. Has the statistical analysis been performed appropriately and rigorously? 

Reviewer #2: Yes

4. Have the authors made all data underlying the findings in their manuscript fully available?

Reviewer #2: Yes

5. Is the manuscript presented in an intelligible fashion and written in standard English?

Reviewer #2: Yes

6. Review Comments to the Author

Reviewer #2: The title includes the term "assessment of face discrimination ability," which may be interpreted as evaluating individual ability—that is, assessing differences between individuals. Since the current manuscript does not establish the tool's reliability for this purpose, I believe the title could be misleading and should be revised to more accurately reflect the paper's main findings.

Apart from this, I believe the paper is ready for publication.

7. PLOS authors have the option to publish the peer review history of their article (what does this mean?). If published, this will include your full peer review and any attached files.

Reviewer #2: No

---

## [Author Response · Author response to Decision Letter 3]

3 Dec 2024

We have adjusted the title to focus on the method and have removed the phrase "assessment of face discrimination ability".

---

## [Editor Report · Decision Letter 4]

5 Dec 2024

A novel, rapid, quantitative method for face discrimination

PONE-D-23-42585R4

Dear Dr. Walter,

We’re pleased to inform you that your manuscript has been judged scientifically suitable for publication and will be formally accepted for publication once it meets all outstanding technical requirements.

Kind regards,

Patrick Bruns

Academic Editor

PLOS ONE
---

## [Editor Report · Acceptance letter]

11 Dec 2024

PONE-D-23-42585R4 

PLOS ONE

Dear Dr. Walter, 

I'm pleased to inform you that your manuscript has been deemed suitable for publication in PLOS ONE. Congratulations! Your manuscript is now being handed over to our production team.

Kind regards, 

on behalf of

Dr. Patrick Bruns 

Academic Editor

PLOS ONE